# Emerging Technologies for Dentin Caries Detection—A Systematic Review and Meta-Analysis

**DOI:** 10.3390/jcm11030674

**Published:** 2022-01-28

**Authors:** Christa Serban, Diana Lungeanu, Sergiu-David Bota, Claudia C. Cotca, Meda Lavinia Negrutiu, Virgil-Florin Duma, Cosmin Sinescu, Emanuela Lidia Craciunescu

**Affiliations:** 1School of Dental Medicine, “Victor Babes” University of Medicine and Pharmacy of Timisoara, 300070 Timisoara, Romania; christa.serban@umft.ro (C.S.); david.bota@umft.ro (S.-D.B.); negrutiu.meda@umft.ro (M.L.N.); sinescu.cosmin@umft.ro (C.S.); petrescu.emanuela@umft.ro (E.L.C.); 2Research Center in Dental Medicine Using Conventional and Alternative Technologies, “Victor Babes” University of Medicine and Pharmacy of Timisoara, 300070 Timisoara, Romania; duma.virgil@osamember.org; 3Center for Modeling Biological Systems and Data Analysis, “Victor Babes” University of Medicine and Pharmacy of Timisoara, 300041 Timisoara, Romania; 4Washington Institute for Dentistry & Laser Surgery, Chevy Chase, MD 20815, USA; author@claudiaccotca.com; 53OM Optomechatronics Group, “Aurel Vlaicu” University of Arad, 310177 Arad, Romania; 6Doctoral School, Polytechnic University of Timisoara, 300222 Timisoara, Romania

**Keywords:** dental caries, non-cavitated dentin caries, diagnosis, occlusal caries, proximal caries, permanent teeth, laser fluorescence, optical coherence tomography (OCT), sensitivity, specificity

## Abstract

This systematic review and meta-analysis aimed at assessing the diagnostic accuracy of emerging technologies, such as laser fluorescence (LF), transillumination, light-emitting diode devices, optical coherence tomography (OCT), alternating current impedance spectroscopy, fluorescence cameras (FC), photo-thermal radiometry, and modulated luminescence technology. In vivo and in vitro results of such non-ionizing, non-invasive, and non-destructive methods’ effectiveness in non-cavitated dentin caries detection are sometimes ambiguous. Following the PRISMA guidelines, 34 relevant research articles published between 2011–2021 were selected. The risk of bias was assessed with a tool tailored for caries diagnostic studies, and subsequent quantitative uni- and bi-variate meta-analysis was carried out in separate sub-groups according to the investigated surface (occlusal/proximal) and study setting (in vivo/in vitro). In spite of the high heterogeneity across the review groups, in vitro studies on LF and FC proved a good diagnostic ability for the occlusal surface, with area under the curve (AUC) of 0.803 (11 studies) and 0.845 (five studies), respectively. OCT studies reported an outstanding performance with an overall AUC = 0.945 (four studies). Promising technologies, such as OCT or FC VistaProof, still need well-designed and well-powered studies to accrue experimental and clinical data for conclusive medical evidence, especially for the proximal surface. Registration: INPLASY202210097.

## 1. Introduction

Dental caries are acknowledged as the most prevalent disease of the oral cavity and affect the vast majority of the population worldwide [1,2]. It is a multifactorial disease characterized by demineralization and loss of tooth structure over time as a result of the interaction between fermentable dietary carbohydrates and acid-producing bacteria [3,4]. Proper detection of dental caries enables clinicians to classify the severity of the lesion and provide optimal therapies that eliminate or reverse decay and preserve healthy tooth structure [5,6]. If identified and treated, dental caries can be prevented from progressing to advanced stages that may lead to tooth loss [7].

Visual and tactile examinations are traditional methods used by dental practitioners to detect caries lesions [8,9]. However, these examination approaches have limitations in their ability to ascertain caries due to their subjective and qualitative nature [10,11].

Dental radiographs are the most common technique for imaging dental caries and dental hard tissue [12]. Despite being a more objective method than the visual examination, radiographic examination may underestimate the caries lesions’ depth [11,13,14]. An emerging technology that has shown potential for dental imaging applications is optical coherence tomography (OCT) [12,15,16,17]. Moreover, it has applications in imaging caries at all stages of progression, and dentin structures, as well [18,19,20,21]. OCT is a non-invasive, non-ionizing, and non-destructive imaging technique that has the potential to facilitate detection of caries and become a valuable tool in clinical practice [22].

Due to the declining caries trends worldwide as a result of fluoride introduction and emphasis on oral hygiene maintenance [23], there is an increased demand for effective methods of detecting caries lesions at an early stage. Caries lesions can be cavitated or non-cavitated. To enable conservative caries management, it is imperative that caries are detected at a non-cavitated stage [5]. Non-cavitated caries lesions (NCCLs) have the surface of enamel still intact but in depth they can reach the enamel or dentin level [8,9]. Throughout the past decades, various new and innovative tools for caries detection have been developed, steadily progressing towards contemporary clinical practice [5]. The shortcomings of traditional visual and radiographic methods to detect NCCLs highlight the need for further investigation of alternative methods for caries detection [8,9,10,24]. Such methods include non-ionizing technologies based on fluorescence, light transillumination, light-emitting diode devices, fiber-optic transillumination, optical coherence tomography, alternating current impedance spectroscopy, optical coherence tomography and photo-thermal radiometry, and modulated luminescence [24].

Previous studies have evaluated the performance of emerging technologies in detecting and quantifying carious lesions [25,26,27,28,29,30], but results from in vivo and in vitro studies have been heterogeneous and there is ambiguity regarding their overall effectiveness as diagnostic methods in clinical practice. Recent systematic reviews have investigated the most commonly used caries detection methods for occlusal and proximal studies, but they either overlooked less studied or reported methods [29,30], such as near-infrared light transillumination and alternating current impedance spectroscopy, or included only in vivo studies [28].

The aim of this systematic review and meta-analysis was to answer the question: what is the diagnostic test accuracy of emerging technologies for non-cavitated (NC) dentin caries detection? This question was answered considering in vivo and in vitro studies that reported results regarding the occlusal and proximal surfaces, over the last 10 years.

## 2. Materials and Methods

This study was performed in accordance with the Preferred Reporting Items for Systematic Reviews and Meta-Analyses (PRISMA) guidelines [31] and “Cochrane Handbook for Diagnostic Test Accuracy Reviews” [32]. The review was constructed based on the PICOS framework [33]: (P) participants—NC dentin caries; (I) intervention—diagnostic tests using emerging technologies for caries detection; (C) comparison—gold standard; (O) outcome—sensitivity, specificity, area under the receiving operating curve (AUC); (S) study design—in vitro and in vivo settings.

The protocol for this systematic review was registered on INPLASY (Unique ID 202210097) and is available in full on the International Platform of Registered Systematic Review and Meta-analysis Protocols (INPLASY) (https://inplasy.com/inplasy-2022-1-0097/, accessed on 19 January 2022).

### 2.1. Eligibility Criteria

Studies eligible for inclusion in the review examined the validity of one or more caries detection technology for diagnosis of NC dentin-level primary caries. Both in vitro and in vivo studies were accepted. Any sample size was accepted. There was no limit on the age of the population under in vivo conditions. Studies analyzing smooth, proximal or occlusal surfaces of human permanent teeth were accepted. The following index tests were accepted: laser fluorescence (LF; DIAGNOdent 2095 or 2190, i.e., “DDpen”, KaVo, Biberach, Germany), fluorescence camera (FC; VistaProof or VistaCam iX, Durr Dental, Bietigheim-Bissingen, Germany), near-infrared light transillumination (NIR-LIT; DIAGNOcam, KaVo, Biberach, Germany), light-emitting diode-based device (LED; Midwest Caries I.D.), optical coherence tomography (OCT), fiber-optic transillumination (FOTI; Electro-Optical Science Inc., Irvington, NY, USA), quantitative light-induced fluorescence (QLF; Inspektor Research Systems BV, Amsterdam, The Netherlands), light-induced fluorescence (LIF; SoproLife, SOPRO, ACTEON Group, La Ciotat, France), alternating current impedance spectroscopy (ACIS; CarieScan PRO, CarieScan LTD, Dundee, Scotland), photo-thermal radiometry and modulated luminescence (PTR-LUM; Canary System, Quantum Dental Technologies, Toronto, Canada). Accepted reference standard tests were: histology, micro-computed tomography (micro-CT), operative validation, and cone-beam computed tomography systems (CBCT). As radiography is commonly used in clinical settings, all types of conventional and digital bitewing radiographs used in conjunction with visual examination were considered as acceptable reference standard tests for in vivo studies. The included studies were published in English within the last decade (January 2011 to August 2021).

### 2.2. Information Sources

Electronic databases of Medline, Embase and PubMed were searched for articles published from 1 January 2011 to 1 September 2021. Medline and Embase databases were searched concomitantly using the Ovid interface. To find articles potentially missed by the search, Google Scholar was queried for diagnostic validity studies pertaining to technologies for caries diagnosis.

### 2.3. Search Strategy

The search was divided into three categories (Table 1). The first category was associated with finding studies related to the clinical situation under investigation so the term “dental caries” was employed. The second category aimed to capture articles related to caries detection technologies so the following terms were used: “lasers” OR “fluorescence” OR fiber optics” OR “optical coherence tomography” OR “light” OR “transillumination” OR “electrical conductivity”. The third category aimed to capture diagnostic validity studies and the following terms were used: “diagnosis” OR “detection” OR “validity”. Each category was connected to the others through the Boolean tool “AND” and the search was limited to the last decade.

### 2.4. Article Selection

Two reviewers independently selected the articles. Disagreements were resolved by consulting and discussing with an experienced researcher. Following de-duplication, the titles and abstracts were screened to ensure they met the eligibility criteria. Irrelevant studies, systematic reviews, conceptual and methodology articles were eliminated. Studies that examined root caries, artificially developed caries, caries in primary teeth, and caries under sealants and around restorations were excluded. After title and abstract screening, the remaining articles underwent full-text reading. Some articles included research on more than one technology, design set-up, or detection surface. Individual studies were searched to ensure that they used accepted reference standard methods and reported values of true positives (TP), false positives (FP), true negatives (TN), and false negatives (FN) related to diagnosis of NC dentin caries, or had enough data to allow derivation of these values. Studies that examined commercially available prototype versions of technologies were excluded.

### 2.5. Data Collection

Two reviewers conducted the data collection. The following information was extracted from the shortlisted articles: type of caries detection technology, setting (in vivo or in vitro), sample size, tooth surface, index test, reference test and outcome data (values of TP, TN, FP, FN, specificity and sensitivity for diagnosis of NC dentin-level caries). In studies with more than one examiner, the values of the first examiner listed were considered. Manufacturer cut-off points for NC dentin-level caries were primarily used in studies that reported multiple cut-off points.

### 2.6. Qualitative Analysis

A risk of bias (RoB) assessment tool for caries diagnostic studies, developed by Kuhnisch et al. [34], was used to assess the included studies’ quality. The tool is based on existing assessment instruments, such as QUADAS-2 [35,36] and Joanna Briggs Institute Reviewers’ Manual [37], but the signaling questions are tailored for the specific methodology of caries diagnostic studies. The tool consists of four domains: (1) selection and spectrum bias, (2) index test, (3) reference test, (4) study flow and data analysis. Within the four domains, there are a total of 16 signaling questions. Two reviewers independently evaluated the studies; any doubts or disagreements were resolved by consensus.

Quality assessment tools indicate the susceptibility to bias and do not categorize studies as of high or low quality [38], therefore the quality appraisal was not applied to exclude studies from the meta-analysis. Adhering to a strict criterion to exclude studies based on bias susceptibility would significantly reduce the number of included studies or may even exclude valuable studies. Furthermore, implementations of study designs that aim to eliminate certain biases are not always feasible and may create limitations in study methods and results [39]. The studies were selected based on the inclusion criteria and the quality assessment would only raise awareness to possible sources of bias and would stimulate discussion about this review’s limitations.

### 2.7. Quantitative Meta-Analysis

Each caries detection and diagnosis technology was put in a separate review sub-group. Based on the number of included studies in each group, there were three types of analyses potentially carried out: (a) descriptive, (b) univariate, and (c) bivariate. For review groups with one study report, only descriptive analysis was performed. For groups with two reports, only descriptive and univariate analyses were performed. For groups with at least three reports, all three types of analyses were conducted (partially for three-study groups and comprehensively for the others).

#### 2.7.1. Descriptive Summary Statistics

Summaries were produced for sensitivity and specificity, with Chi-square statistical tests for the significance of observed differences. Forest plots and summary receiver operating characteristic (sROC) curves were employed as descriptive graphs. ROC curves illustrate the methods’ diagnostic ability, plotting the sensitivity (i.e., true positive rate) against the false positive rate (FPR, equal to 1-specificity). ROC ellipses plots were drawn for each technology to illustrate the individual studies’ 95% confidence regions on the ROC space.

#### 2.7.2. Univariate Meta-Analysis

The natural logarithm values of decision odds ratios (log DORs) were estimated applying the random-effects model of DerSimonian and Laird [40,41]. Continuity correction of 0.5 was applied to zero cells. Log DOR was the statistic of choice for describing the discrimination ability of diagnostic tests for reasons of consistency across the uni- and bivariate meta-analytical methods. For this statistic, zero is the reference value equivalent to non-discrimination ability of the diagnostic test: the larger the positive log DOR value, the better the diagnostic test. Forest plots for log DORs were used as graphical representation.

Higgins’ I-square, the ratio of true heterogeneity to total variation in observed effects, was employed to analyze the heterogeneity of published results about each index test included in the meta-analysis. I-square was preferred as it is a kind of signal to noise ratio and it is not sensitive to the effect size or the number of studies [41].

Correlation was determined between sensitivity and FPR, as a measure of the overall quality of classification (high values indicate a cut-off threshold effect).

Publication bias was visually assessed with funnel plots of log DORs, which display the relationship between study size and effect size. They offer the advantage of showing the pattern of studies clustering around the mean effect, with apparent asymmetry as evidence of bias [41]. Deeks’ approach [42] was also considered for this analysis, but the small number of identified studies hampered the effectiveness of this procedure for assessing the likely impact of publication bias.

#### 2.7.3. Bivariate Meta-Analysis

The bivariate analysis followed the linear mixed model for diagnostic reviews of Reitsma et al. [43,44]. Summary ROC (sROC) curves based on the bivariate random-effects model fitted by the method of restricted maximum likelihood (REML) were plotted, with 95% confidence prediction regions for the estimate. Aggregated statistics were estimated: area under the curve (AUC), partial AUC (pAUC, restricted to observed FPRs and normalized), sensitivity, and FPR. In this bivariate analysis, continuity correction of 0.5 was applied to all cells.

#### 2.7.4. General Specifications

Meta-analysis was separately conducted for each technology, and each index test, tooth surface and study setting. Confidence intervals of sensitivity, and specificity or FPR (equal to 1-specificity) were used as indicators of diagnostic accuracy for each study.

All analysis was conducted with a confidence level and statistical significance of 0.95 and 0.05, respectively.

Meta-analysis was performed with the R 4.0.5 language and environment for statistical processing (R Core Team, 2021), including packages “mada” (version 0.5.8), “meta” (version 4.18-2), “metafor” (version 3.0-2), and “mvmeta” (version 1.0.3).

## 3. Results

### 3.1. Study Selection and Data Collection

Figure 1 shows the selection flow diagram, according to the PRISMA guidelines. A total of 1729 articles were identified when the search criteria were applied. There were 1132 articles from PubMed, 594 articles from EMBASE and Medline found using the Ovid interface, and 3 articles found through a manual search of Google Scholar. After the removal of 419 duplicated articles, 1310 unique articles remained and underwent title and abstract screening; 1240 articles were further eliminated during this process. The remaining 70 articles underwent full-text reading and 36 articles were excluded for using unacceptable reference standard methods, for not being related to NC dentin caries diagnosis, for being outside the subject area, or for not having available outcome data. A total of 34 articles were found to be suitable for inclusion in the systematic review and meta-analysis. There were 21 articles that studied more than one caries detection technology.

Of the 34 articles included, 21 used LF technology [45,46,47,48,49,50,51,52,53,54,55,56,57,58,59,60,61,62,63,64,65], 9 used FC technology [45,54,55,58,63,64,66,67,68], 4 used NIR-LT technology [53,56,61,69], 5 used LED technology [46,53,60,63,70], 4 used OCT technology [71,72,73,74], 3 used FOTI technology [71,75,76], 4 used QLF technology [48,71,77,78], 2 used LIF technology [62,71], 2 used ACIS technology [56,59], and 1 used PTR-LUM technology [78]. There were more data sets from in vitro studies (*n* = 26) than in vivo studies (*n* = 9). There was one study [65] that produced data sets from both in vitro and in vivo investigations.

The actual data collected from the 34 articles are provided as Appendix A.

### 3.2. Qualitative Analysis

The results from the RoB assessment for each study are illustrated in Figure 2a,b for in vivo and in vitro studies, respectively. The RoB assessment for each study is provided as Appendix A.

### 3.3. Quantitative Meta-Analysis

The univariate and bivariate analysis results for each technology are presented in the following. Supplementary descriptive summary statistics for each technology and each study are also provided (Appendix A).

#### 3.3.1. Laser Fluorescence (DIAGNOdent 2095 and 2190)

The DIAGNOdent 2095 (DD 2095) was assessed in five reports. Meta-analytical results are presented in Table 2. Appendix A (the ROC ellipses plot) corroborated the I-square values, indicating moderate to low heterogeneity. Under in vitro conditions, the DD 2095 had moderate pooled sensitivity and a low false positive rate (i.e., high specificity).

The DIAGNOdent pen (DD pen) was assessed in 19 reports, 11 studies reporting in vitro results for occlusal surface. Table 3 presents the results.

The heterogeneity for the DD pen in vitro occlusal group was moderate (I-square = 42.32%) and comprehensive uni- and bi-variate analysis was conducted: moderate pooled sensitivity and specificity resulted. The log DOR forest and funnel plots in Figure 3 illustrate the DD pen diagnostic performance and publication bias: the results are quite homogeneous, with only one outlier. The sROC curve resulted in AUC = 0.803 and pAUC = 0.702 (i.e., close to AUC). Figure 4 illustrates the sROC and the 95% confidence region of the aggregate bi-dimensional estimate, suggesting moderate and reliable diagnostic performance.

#### 3.3.2. Fluorescence Camera (VistaCam iX and VistaProof)

There were 12 studies that assessed the VistaCam iX and VistaProof (Table 4). Most of them (*n* = 8) investigated the VistaProof in vitro. Considerably higher sensitivity resulted when optimal cut-off was applied, compared to the diagnostic performance when the manufacturer cut-off values were used. In the former case, the penalty in specificity was rather small, with FPR of 0.33, 95% CI (0.27; 0.38) compared to 0.08, 95%CI (0.03; 0.21) for the latter. The low I-square values indicated low heterogeneity.

Comprehensive meta-analysis was conducted on the five-study group of in vitro VistaProof using optimal cut-off. The log DOR forest and funnel plots are illustrated in Figure 3a,b: they display homogeneity, as I-squared = 26.32% also suggested. The sROC for this index test (Figure 4) illustrates the high values of sensitivity, with good AUC = 0.845 and pAUC = 0.871.

#### 3.3.3. Optical Coherence Tomography (OCT)

The OCT was assessed in only four reports, as Table 5 shows. Nevertheless, since it provides 2D and 3D images of occlusal and proximal caries, we conducted a supplementary combined meta-analysis on all four studies, irrespective of their setting or surface.

Although I-squared was relatively high, the forest and funnel plots in Figure 3 displayed reliable results. Moreover, the sROC in Figure 4 illustrated the very good diagnostic ability of this technology, with AUC = 0.945 and pAUC = 0.836. The conspicuous outlier was the study of Gomez et al. 2013 [71], an occlusal in vitro study with RoB issues related to teeth, caries and sample selection, and the tests’ calibration (supplemental details regarding the actual data are shown in Appendix A).

#### 3.3.4. Near-Infrared Light Transillumination (NIR-LT, DIAGNOcam)

The results are presented in Table 6: only one or two studies in each group allowed little more than simple descriptive statistics.

In addition to these results, the Appendix A would illustrate the high heterogeneity of this technology’s results (the ROC ellipses plot in Appendix A).

#### 3.3.5. Light-Emitting Diode-Based Device (LED Device, MIDWEST)

Results for this technology are presented in Table 7.

As in the previous section, Appendix A (the ROC ellipses plot in Appendix A; they can be observed in addition to the results included in the main text; these diagnostic tests had less heterogeneity, but displayed low diagnostic performance).

#### 3.3.6. Fibre-Optic Transillumination (FOTI)

Results for this technology are presented in Table 8.

#### 3.3.7. Quantitative Light-Induced Fluorescence (QLF)

Results for this technology are presented in Table 9. For this technology we could identify only in vitro results.

Appendix A (namely, the ROC ellipses plot in Appendix A) would corroborate the good bivariate meta-analytical results for occlusal surface in Table 9.

#### 3.3.8. Light-Induced Fluorescence (LIF)

Results for this technology are presented in Table 10.

#### 3.3.9. Alternating Current Impedance Spectroscopy (ACIS, CarieScan PRO)

Results for this technology are presented in Table 11.

#### 3.3.10. Photo-Thermal Radiometry and Modulated Luminescence (PTR-LUM)

Results for this technology (only one study) are presented in Table 12.

## 4. Discussion

Most of the previous systematic reviews and meta-analyses about technologies in caries detection have focused on traditional and commonly used non-traditional technologies [25,26,27,29,30]. They did not address or study new methods, such as the Midwest Caries I.D. and SoproLife. Although Gomez et al. [79] studied a wide range of emerging methods, since its publication many more advances in technology have occurred, such as the release of the Canary System, and new original studies have been published. More recently, Foros et al. [28] published a review on detection methods for early caries, but only in vivo studies were considered. In order to gain a comprehensive view and address RoB in both in vitro and in vivo studies, data from both clinical and laboratory settings were used in the present meta-analysis. On the other hand, since the current review addressed the whole range of emerging methods (that is to say, it included the less common and less investigated ones), only a limited number of studies and high heterogeneity could be found.

### 4.1. Laser Fluorescence Technology

DIAGNOdent (DD) groups, utilizing the LF technology, comprised the largest number of studies included in this meta-analysis. There are two DD devices: DIAGNOdent 2095 (DD 2095) and DIAGNOdent pen 2190 (DD pen) [28,80]. DD 2095 works only on occlusal surfaces, whereas the DD pen works on both occlusal and proximal surfaces [81,82,83]. This review included studies published within the past decade and most of them assessed the DD pen diagnosis performance. There were fewer studies about the DD 2095, thus suggesting its use and perceived interest has declined in the past 10 years since the release of the DD pen.

DD 2095 was assessed in five reports. Under experimental in vitro conditions, the DD 2095 had moderate to low pooled sensitivity associated with rather high specificity (i.e., low FPR). However, the few identified studies made it difficult to conclude on the actual performance of the device.

DD pen was assessed in 19 reports. Most studies were carried out under in vitro conditions and evaluated occlusal surfaces. The pooled sensitivity was high and the FPR was low, implying high specificity. The publication bias of in vitro DD pen occlusal studies was low, with most studies fitting into the funnel and only one outlier. There was moderate heterogeneity among the 11 in vitro occlusal studies, with high area under the sROC curve, thus proving good discriminative capacity. The overall good performance of the DD pen that resulted from this systematic review and meta-analysis was in accordance with previous studies concerning experimental in vitro conditions [26,27,28].

Fewer in vivo DD pen studies evaluating occlusal surfaces met the inclusion criteria and indicated the more difficult nature of in vivo studies designs, due to limitations in the ability to accurately validate all types of caries and teeth [34]. The included in vivo studies of occlusal surfaces using a DD pen exhibited higher heterogeneity, probably generated by the variations in sample size and difficulties in obtaining a true representative sample. This effect was also apparent in the qualitative RoB assessment, particularly in domains 1 and 3 (concerned with selection and spectrum bias, and reference test bias, respectively).

Studies carried out on occlusal surfaces using a DD pen, in both in vivo and in vitro conditions, exhibited lower heterogeneity than studies carried out on proximal surfaces. A low number of reports were identified for proximal NC dentin caries under in vitro conditions and they reported a high pooled specificity (i.e., low FPR) but inconclusive sensitivity. Proximal caries are more difficult to diagnose [84,85], therefore the high heterogeneity in proximal surface sub-groups may be due to the variations in reference standards and their (in)ability to detect proximal caries. Although the DD pen is able to detect proximal caries, it is more challenging to conduct such an investigation under in vivo conditions when proximal tooth contacts are tight; it is difficult to accurately simulate this in vitro, as well [81]. DD pen tips have a minimal thickness of 0.4 mm and tooth separation is necessary to increase accessibility for the DD pen tip, especially in tight contact areas [81]. Due to these additional challenges of proximal caries detection with the DD pen, fewer studies were identified.

Differences in the experimental designs (i.e., simulating proximal caries in vitro) and in separating tight proximal contacts between teeth in clinical in vivo set-ups may have introduced additional heterogeneity. For example, in case of DD pen for proximal NC dentin caries, in vivo studies used operative validation as the reference standard [53,61], and one study in the in vitro sub-group used visual examination and bitewing radiographs as the reference standard [47]. This less rigorous validation may also bring limitations and introduce heterogeneity [34].

Summing up, the DD pen proved to have good diagnostic performance in experimental in vitro studies for occlusal NC dentin caries, but there is inconclusive evidence regarding the NC dentin caries detection on proximal surfaces.

### 4.2. Fluorescence Camera Technology

There were 12 reports that assessed the FC VistaProof and VistaCam iX instruments, all on occlusal NC dentin caries detection. Considerably higher sensitivity was observed when optimal cut-off was used (maintaining reasonably moderate specificity), compared to the manufacturer cut-off values. Low-sensitivity results in low negative predictive values, which might be an issue when manufacturer cut-off values are used in clinical practice. Therefore, these results strongly suggest a reconsideration of the manufacturer’s cut-off values.

In addition, the small (i.e., tight) 95% confidence interval of the pooled sensitivity for in vitro VistaProof instruments indicates good stability and reliable estimation of performance when optimal cut-offs are employed. The sROC curve also showed excellent sensitivity in case of optimal cut-off values.

On the other hand, the low number of in vivo studies that met the inclusion criteria restrains the conclusions on this technology’s efficacy under clinical conditions.

### 4.3. Optical Coherence Tomography

OCT appears to be a promising tool as high sensitivity and specificity were observed in three out of four individual datasets. Since OCT is an imaging technology that can provide accurate 2D and 3D visualization of caries lesions for both occlusal and proximal surfaces, by using the same technical principles [86,87,88], the datasets of the four identified OCT studies that met the inclusion criteria were combined for an overall meta-analysis. The aggregated results confirmed high sensitivity and specificity. Even though the comprehensive meta-analysis included results on different surfaces and design settings, only moderate to high heterogeneity was observed. The sROC curve clearly showed this outstanding discriminative capacity, in spite of the large 95% confidence area on the ROC space (generated by the outlier study).

These findings suggest OCT as a promising emerging technology, although more studies are needed to confirm these results and further assessment of this technology should be carried out.

### 4.4. Newly Emerging Technologies

The remaining emerging technologies in this review had few identified studies that met the inclusion criteria. Although recent systematic reviews and meta-analyses overlooked these technologies probably due to the low number of available studies [29,30], their diagnostic potential should be acknowledged. We systematically identified these studies, collected the performance data, synthesized them and then put them in context, irrespective of the reports’ number; this contribution might help future investigators to build up on existing body of evidence. For example, previous research considered the QLF as a good diagnostic instrument [89] and this review’s results (out of three identified studies) found consistency in the NC dentin caries detection performance, suggested by the overlap in the confidence intervals on the ROC ellipse plot; it was further confirmed by the high AUC value, and high pooled sensitivity and specificity.

Nevertheless, additional assessment of FOTI, QLF, LIF, ACIS, and PTR-LUM technologies is needed to determine their diagnostic performance in occlusal and proximal NC caries detection using the RoB criteria set out by Kuhnisch et al. [34]. The RoB assessment tool was useful in the identification and synthesis of potential sources of bias. Many in vivo and in vitro studies exhibited high RoB in items from domain 1, concerned with selection and spectrum bias. Bias and heterogeneity of the included studies may be due to the challenges associated to enrolling a population-based, randomized sample of patients, teeth, and NC dentin caries. In addition, it is important to note the difference in the potential sources of bias between in vitro and in vivo studies in terms of the reference tests. Higher RoB was found in the in vivo studies across the domain 3 items, generated by the inability to apply rigorous validation methods under clinical conditions (due to the destructive nature of such gold standards). Furthermore, RoB was noted for the calibration of index and reference tests, as many studies did not specify the extent of their investigators’ training and calibration.

### 4.5. Strengths and Limitations

A strong methodological point was the inclusion of all emerging diagnostic methods for proximal and occlusal NC dentin caries detection into one comprehensive systematic review and meta-analysis, which filled the gaps in the currently available literature. This approach also allowed for a full assessment of the technologies and their comparison.

Limitations of this review are mainly due to the low number of studies available for some sub-groups and technologies. This generated caveats regarding the generalization of the meta-analytical results. Supplementary limitation is that studies with high and moderate RoB were included together in the meta-analysis, resulting in considerable heterogeneity. Nonetheless, the review acknowledged the heterogeneity and RoB, and identified the areas where future, well-designed studies and research are in great need.

## 5. Conclusions

The present systematic review extended the methodological search for published results on emerging technologies for NC dentin caries detection into comprehensive data collection and systematic synthesis, irrespective of the studies’ scarcity, a paucity that is especially prominent for proximal NCCLs. The review confirmed the many emerging technologies aiming at an accurate detection of NCCLs.

The qualitative and quantitative analysis would help identify gaps in the available data and evidence about these diagnostic methods. Being the most studied emerging tool, LF is an acceptable technology in detecting NC dentin caries; however, when applied in clinical practice and for proximal NCCLs in particular, it must be used in conjunction with conventional methods to ensure good diagnostic test accuracy.

Promising technologies, such as OCT, FC VistaProof, or QLF still need well-designed and well-powered studies to accrue experimental and clinical data for conclusive medical evidence. Nevertheless, technologies that enable imaging of the caries lesions with non-ionizing radiation, such as OCT, have the potential to enable more accurate detection of occlusal NCCLs compared to radiographs, and to offer the missing dependable diagnostic instrument for proximal NC dentin caries. Provided further development and investigations are carried out, such technologies are foreseen as becoming valuable tools in actual clinical practice.

This review calls to experts and professional bodies to develop, and agree upon, methodologies for assessing the studies’ quality, level of evidence, and ethical requirements in employing and comparing different technologies for dental caries detection and diagnosis.

## Figures and Tables

**Figure 1 jcm-11-00674-f001:**
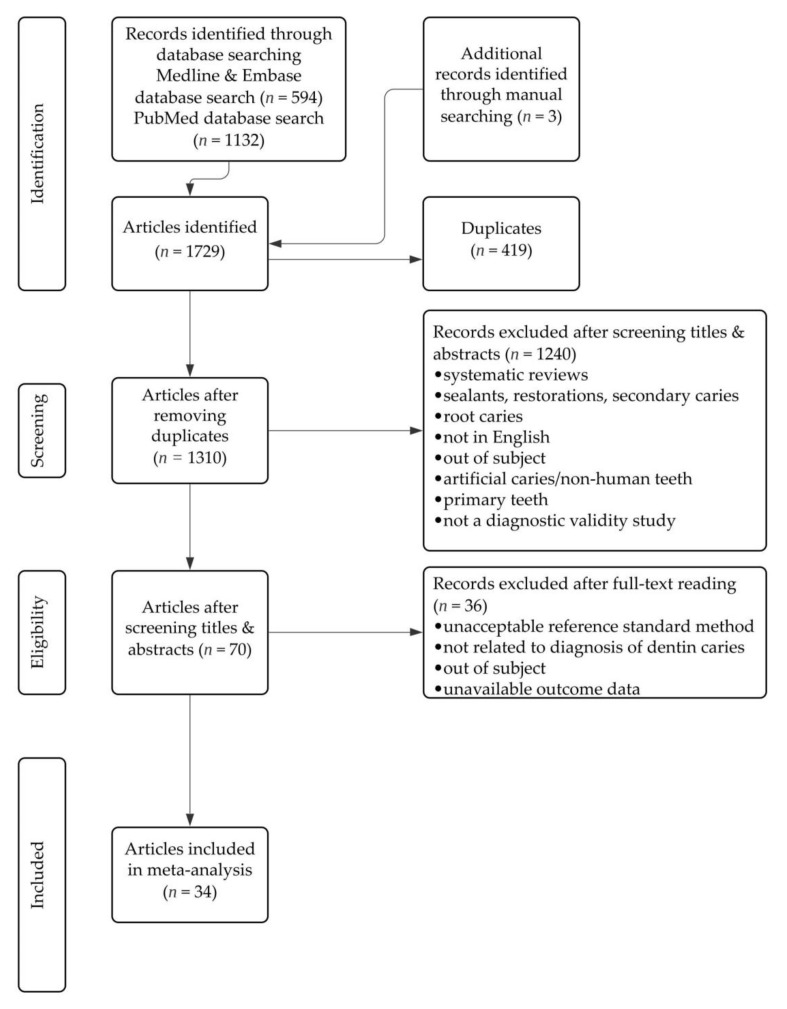
Flow diagram of the study selection process.

**Figure 2 jcm-11-00674-f002:**
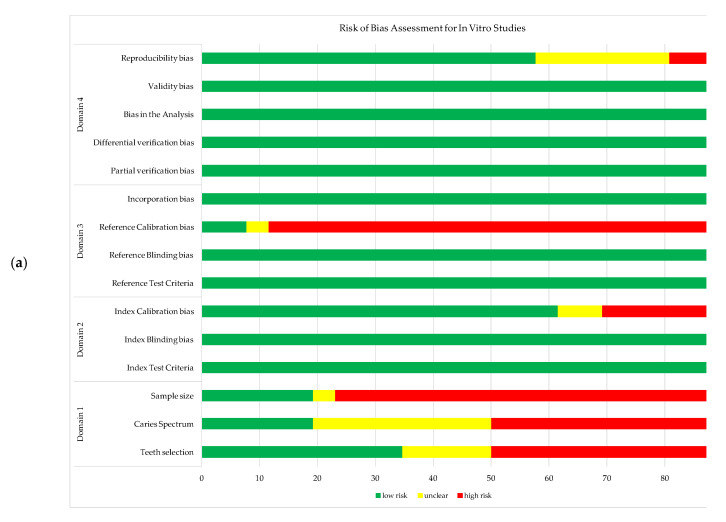
Risk of bias assessment for the studies include in meta-analysis: (**a**) in vivo studies; (**b**) in vitro studies.

**Figure 3 jcm-11-00674-f003:**
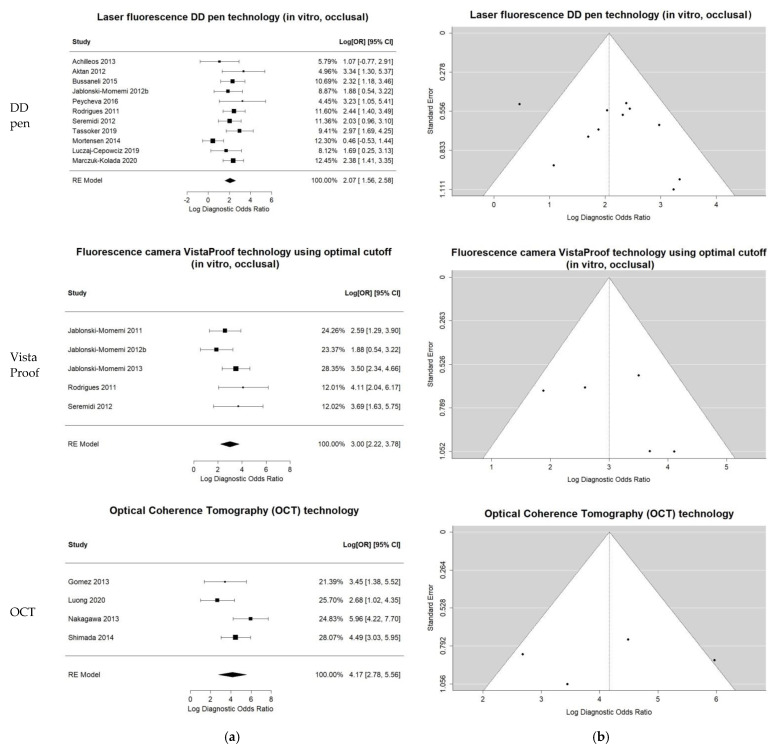
Univariate summary plots for the three groups of index tests included in the comprehensive bivariate meta-analysis: LF DD pen occlusal, in vitro; FC VistaProof using optimal cut-off, in vitro; OCT overall. (**a**) Log DOR forest plots and random effects aggregate estimates. (**b**) Log DOR funnel plots to illustrate the publication bias as patterns of individual studies clustering around the mean effect.

**Figure 4 jcm-11-00674-f004:**
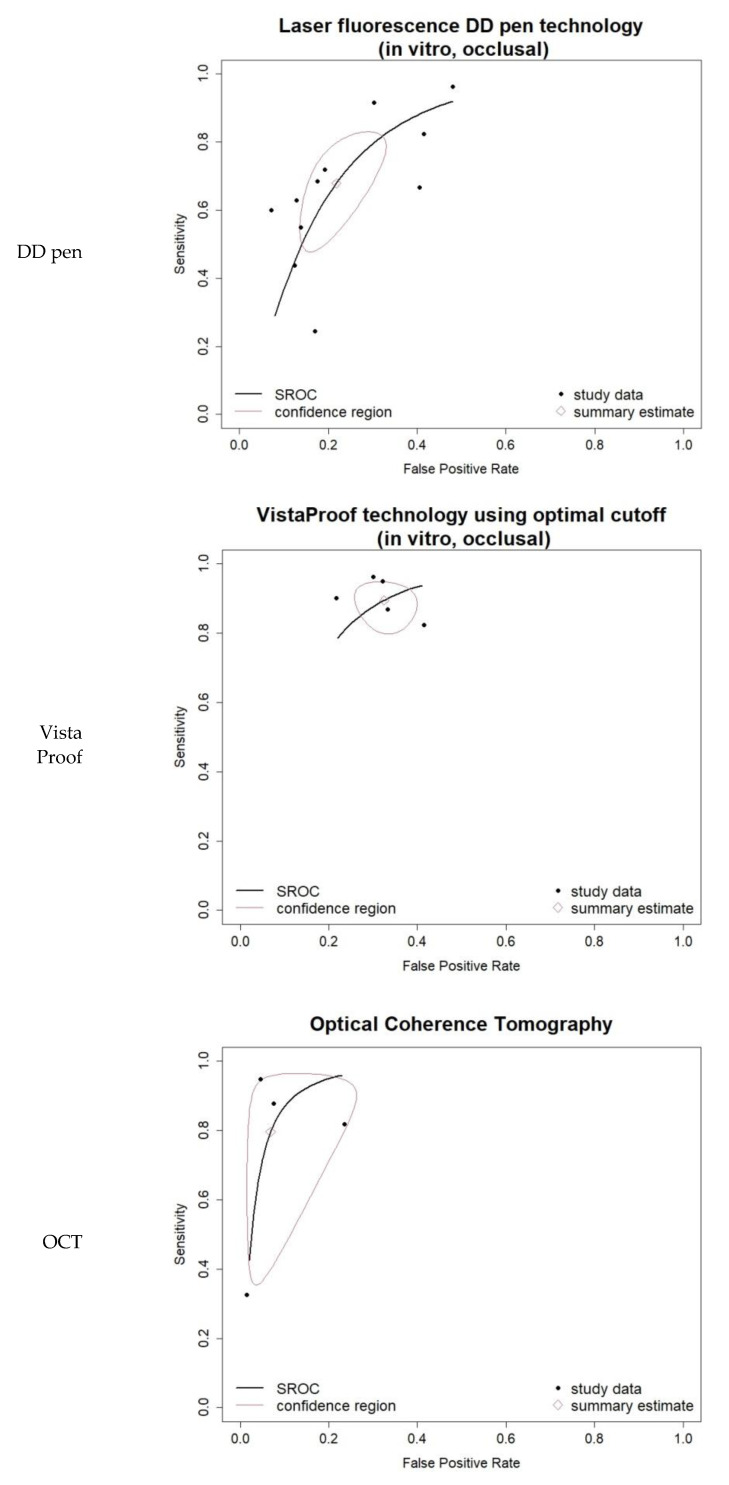
Bivariate summary ROC (sROC) curves for the three groups of index tests included in the comprehensive bivariate meta-analysis: LF DD pen occlusal, in vitro; FC VistaProof using optimal cut-off, in vitro; OCT overall. They also integrate the individual studies’ data in a summary bi-dimensional estimate and show its 95% prediction region.

**Table 1 jcm-11-00674-t001:** Search strategy used to search PubMed, EMBASE, and Medline.

Search Category	Search Items
1.	“dental caries”
	AND
2.	“lasers” OR “fluorescence” OR fiber optics” OR optical coherence tomography” OR “light” OR “transillumination” OR “electrical conductivity”
	AND
3.	“diagnosis” OR “detection” OR “validity”.

**Table 2 jcm-11-00674-t002:** Meta-analytical results for the DIAGNOdent 2095 test. Only occlusal studies were identified and included in this analysis.

Laser Fluorescence (LF)			
Index Test	In Vivo/ In Vitro	Meta-Analytical Statistics	Occlusal
DIAGNOdent 2095	In vivo	N	2
		Univariate	
		I-square [%]	0%
		log DOR ± se; 95%CI	2.79 ± 0.53 (1.75; 3.83)
		R; 95%CI	–
DIAGNOdent 2095	In vitro	N	3
		Univariate	
		I-square [%]	43.29%
		log DOR ± se; 95%CI	1.74 ± 0.44 (0.89; 2.6)
		R; 95%CI	–
		Bivariate	
		Tsens ± se; 95%CI	0.14 ± 0.86 (−0.44; 0.71)
		Tfpr ± se; 95%CI	−1.61 ± 0.27 (−2.13; −1.09)
		Sensitivity; 95%CI	0.53 (0.39; 0.67)
		FPR; 95%CI	0.17 (0.11; 0.25)
		AUC	0.745
		pAUC	0.559

Abbreviations: AUC, area under the curve; CI, confidence interval; DOR, diagnostic odds ratio; FPR, false positive rate; pAUC, partial area under the curve; R, correlation coefficient between sensitivity and false positive rate; se, standard error of the estimate; Tfpr, log-transformed false positive rate; Tsens, log-transformed sensitivity.

**Table 3 jcm-11-00674-t003:** Meta-analytical results for the DD pen test. The sub-group of 11 studies reporting occlusal in vitro results, with I-squared = 42.32%, underwent comprehensive meta-analysis (including graphs).

Laser Fluorescence (LF)				
Index Test	In Vivo/ In Vitro	Meta-Analytical Statistics	Occlusal	Proximal
DD pen	In vivo	N	3	2
		Univariate		
		I-square [%]	64.61%	92.30%
		log DOR ± se; 95%CI	2.24 ± 0.54 (1.19; 3.3)	1.27 ± 2.09 (−2.82; 5.36)
		R; 95%CI	–	–
		Bivariate		
		Tsens ± se; 95%CI	0.95 ± 0.86 (−0.05; 1.94)	–
		Tfpr ± se; 95%CI	−1.24 ± 0.25 (−1.72; −0.75)	–
		Sensitivity; 95%CI	0.72 (0.49; 0.88)	–
		FPR; 95%CI	0.23 (0.15; 0.32)	–
		AUC	0.811	–
		pAUC	0.672	–
DD pen	In vitro	N	11	3
		Univariate		
		I-square [%]	42.32%	92.81%
		log DOR ± se; 95%CI	2.07 ± 0.26 (1.56; 2.58)	3.67 ± 1.2 (1.32; 6.02)
		R; 95%CI	0.67 (0.12; 0.91)	–
		Bivariate		
		Tsens ± se; 95%CI	0.75 ± 0.34 (0.08; 1.42)	1.17 ± 0.86 (−0.52; 2.86)
		Tfpr ± se; 95%CI	−1.28 ± 0.23 (−1.74; −0.82)	−2.4 ± 0.49 (−3.36; −1.44)
		Sensitivity; 95%CI	0.68 (0.52; 0.81)	0.76 (0.37; 0.95)
		FPR; 95%CI	0.22 (0.15; 0.31)	0.08 (0.03; 0.19)
		AUC	0.803	0.932
		pAUC	0.702	0.743

Abbreviations: AUC, area under the curve; CI, confidence interval; DOR, diagnostic odds ratio; FPR, false positive rate; pAUC, partial area under the curve; R, correlation coefficient between sensitivity and false positive rate; se, standard error of the estimate; Tfpr, log-transformed false positive rate; Tsens, log-transformed sensitivity.

**Table 4 jcm-11-00674-t004:** Meta-analytical results for FC technology. The five-study sub-group reporting in vitro results by using optimal cut-off values for the VistaProof test, with I-squared = 26.32%, underwent comprehensive meta-analysis (including graphs).

Fluorescence Camera (FC)				
Index Test	In Vivo/ in Vitro	Meta-Analytical Statistics	Occlusal Using Manufacturer Cut-Off	Occlusal Using Optimal Cut-Off
VistaProof	In vivo	N	1	–
		Univariate		
		I-square [%]	–	–
		log DOR; 95%CI	2.73 (1.50; 3.96)	–
		R; 95%CI	–	–
VistaProof	In vitro	N	3	5
		Univariate		
		I-square [%]	0%	26.32%
		log DOR ± se; 95%CI	1.014 ± 0.63 (−0.22; 2.24)	3.00 ± 0.40 (2.22; 3.78)
		R; 95%CI	–	−0.55 (−0.97; 0.64)
		Bivariate		
		Tsens ± se; 95%CI	−1.63 ± 0.94 (−3.46; 0.21)	2.14 ± 0.31 (1.53; 2.76)
		Tfpr ± se; 95%CI	−2.51 ± 0.59 (−3.66; −1.36)	−0.73 ± 0.13 (−0.99; −0.47)
		Sensitivity; 95%CI	0.16 (0.03; 0.55)	0.895 (0.82; 0.94)
		FPR; 95%CI	0.08 (0.03; 0.21)	0.33 (0.27; 0.38)
		AUC	0.762	0.845
		pAUC	0.253	0.871
VistaCam iX	In vitro	N	2	1
		Univariate		
		I-square [%]	0%	–
		log DOR ± se; 95%CI	2.35 ± 0.42 (1.54; 3.16)	2.56 (1.56; 3.57)
		R; 95%CI	–	–

Abbreviations: AUC, area under the curve; CI, confidence interval; DOR, diagnostic odds ratio; FPR, false positive rate; pAUC, partial area under the curve; R, correlation coefficient between sensitivity and false positive rate; se, standard error of the estimate; Tfpr, log-transformed false positive rate; Tsens, log-transformed sensitivity.

**Table 5 jcm-11-00674-t005:** Meta-analytical results for the optical coherence tomography (OCT) test. For this technology, supplemental comprehensive meta-analysis was conducted on the four identified studies (including graphs), irrespective of their review sub-group.

Optical Coherence Tomography (OCT)				
Index Test	In Vivo/ In Vitro	Meta-Analytical Statistics	Occlusal	Proximal
OCT	In vivo	N	–	1
		Univariate		
		I-square [%]	–	–
		log DOR; 95%CI	–	4.49 (3.03; 5.95)
		R; 95%CI	–	–
OCT	In vitro	N	2	1
		Univariate		
		I-square [%]	0%	–
		log DOR ± se; 95%CI	2.98 ± 0.66 (1.69; 4.28)	5.96 (4.22; 7.70)
		R; 95%CI	–	–
OCT overall	In vivo		Occlusal & proximal	
	& in vitro	N	4	
		Univariate		
		I-square [%]	61.87%	
		log DOR ± se; 95%CI	4.17 ± 0.71 (2.78; 5.56)	
		R; 95%CI	0.37 (−0.92; 0.98)	
		Bivariate		
		Tsens ± se; 95%CI	1.36 ± 0.80 (−0.21; 2.94)	
		Tfpr ± se; 95%CI	−2.61 ± 0.64 (−3.87; −1.35)	
		Sensitivity; 95%CI	0.80 (0.45; 0.95)	
		FPR; 95%CI	0.07 (0.02; 0.21)	
		AUC	0.945	
		pAUC	0.836	

Abbreviations: AUC, area under the curve; CI, confidence interval; DOR, diagnostic odds ratio; FPR, false positive rate; pAUC, partial area under the curve; R, correlation coefficient between sensitivity and false positive rate; se, standard error of the estimate; Tfpr, log-transformed false positive rate; Tsens, log-transformed sensitivity.

**Table 6 jcm-11-00674-t006:** Meta-analytical results for the near-infrared light transillumination (DIAGNOcam) test. Only univariate statistics could be determined, due to the limited number of studies in each sub-group.

Near-InfraRed Light Transillumination (NIR-LT)				
Index Test	In Vivo/ In Vitro	Meta-Analytical Statistics	Occlusal	Proximal
DIAGNOcam	In vivo	N	1	2
		Univariate		
		I-square [%]	–	93.66%
		log DOR; 95%CI	4.16 (2.10; 6.22)	3.56 ± 2.44 (−1.22; 8.34)
		R; 95%CI	–	–
DIAGNOcam	In vitro	N	–	1
		Univariate		
		I-square [%]	–	–
		log DOR; 95%CI	–	5.46 (3.59; 7.34)
		R; 95%CI	–	–

Abbreviations: AUC, area under the curve; CI, confidence interval; DOR, diagnostic odds ratio; FPR, false positive rate; pAUC, partial area under the curve; R, correlation coefficient between sensitivity and false positive rate; se, standard error of the estimate.

**Table 7 jcm-11-00674-t007:** Meta-analytical results for the Midwest test. Only univariate statistics could be determined, due to the limited number of studies in each sub-group.

Light-Emitting Diode-Based Device (LED)				
Index Test	In Vivo/ In Vitro	Meta-Analytical Statistics	Occlusal	Proximal
Midwest	In vivo	N	–	1
		Univariate		
		I-square [%]	–	–
		log DOR; 95%CI	–	1.71 (1.29; 2.13)
		R; 95%CI	–	–
Midwest	In vitro	N	2	2
		Univariate		
		I-square [%]	49.44%	25.59%
		log DOR ± se; 95%CI	2.48 ± 0.53 (1.44; 3.52)	1.67 ± 0.54 (0.62; 2.72)
		R; 95%CI	–	–

Abbreviations: AUC, area under the curve; CI, confidence interval; DOR, diagnostic odds ratio; FPR, false positive rate; pAUC, partial area under the curve; R, correlation coefficient between sensitivity and false positive rate; se, standard error of the estimate.

**Table 8 jcm-11-00674-t008:** Meta-analytical results for the fibre-optic transillumination (FOTI) test. Only descriptive statistics could be determined, each for the single study in each sub-group.

Fiber-Optic Transillumination (FOTI)				
Index Test	In Vivo/ In Vitro	Meta-Analytical Statistics	Occlusal	Proximal
FOTI	In vivo	N	–	1
		Univariate		
		I-square [%]	–	–
		log DOR; 95%CI	–	2.46 (2.06; 2.87)
		R; 95%CI	–	–
FOTI	In vitro	N	1	1
		Univariate		
		I-square [%]	–	–
		log DOR; 95%CI	3.78 (2.63; 4.93)	2.28 (0.81; 3.75)
		R; 95%CI	–	–

Abbreviations: CI, confidence interval; DOR, diagnostic odds ratio.

**Table 9 jcm-11-00674-t009:** Meta-analytical results for the quantitative light-induced fluorescence (QLF) Inspektor Pro test.

Quantitative Light-Induced Fluorescence (QLF)				
Index Test	In Vivo/ In Vitro	Meta-Analytical Statistics	Occlusal	Proximal
QLF Inspektor Pro	In vitro	N	3	1
		Univariate		
		I-square [%]	0%	–
		log DOR ± se; 95%CI	2.75 ± 0.31 (2.15; 3.35)	2.55(1.27; 3.83)
		R; 95%CI	–	–
		Bivariate		
		Tsens ± se; 95%CI	1.66 ± 0.39 (0.90; 2.41)	–
		Tfpr ± se; 95%CI	−1.18 ± 0.34 (−1.85; −0.51)	–
		Sensitivity; 95%CI	0.84 (0.71; 0.92)	–
		FPR; 95%CI	0.24 (0.14; 0.38)	–
		AUC	0.873	–
		pAUC	0.854	–

Abbreviations: AUC, area under the curve; CI, confidence interval; DOR, diagnostic odds ratio; FPR, false positive rate; pAUC, partial area under the curve; R, correlation coefficient between sensitivity and false positive rate; se, standard error of the estimate; Tfpr, log-transformed false positive rate; Tsens, log-transformed sensitivity.

**Table 10 jcm-11-00674-t010:** Meta-analytical results for the LIF test. Only partial univariate statistics could be determined, due to the limited number of studies.

Light-Induced Fluorescence (LIF)			
Index Test	In Vivo/ In Vitro	Meta-Analytical Statistics	Occlusal
LIF	In vitro	N	2
		Univariate	
		I-square [%]	0%
		log DOR ± se; 95%CI	3.52 ± 0.50 (2.54; 4.50)
		R; 95%CI	–

Abbreviations: CI, confidence interval; DOR, diagnostic odds ratio; se, standard error of the estimate.

**Table 11 jcm-11-00674-t011:** Descriptive results for the alternating current impedance spectroscopy (ACIS) technology, CarieScan PRO test.

Alternating Current Impedance Spectroscopy (ACIS)			
Index Test	In Vivo/ In Vitro	Meta-Analytical Statistics	Occlusal
CarieScan PRO	In vivo	N	1
		Univariate	
		I-square [%]	–
		log DOR; 95%CI	3.56 (2.20; 4.91)
		R; 95%CI	–
CarieScan PRO	In vitro	N	1
		Univariate	
		I-square [%]	–
		log DOR; 95%CI	1.45 (−0.99; 3.89)
		R; 95%CI	–

Abbreviations: CI, confidence interval; DOR, diagnostic odds ratio.

**Table 12 jcm-11-00674-t012:** Descriptive results for the photo-thermal radiometry and modulated luminescence (PTR-LUM) technology, Canary System test.

PhotoThermal Radiometry and Modulated Luminescence (PTR-LUM)			
Index Test	In Vivo/ In Vitro	Meta-Analytical Statistics	Occlusal
Canary System	In vitro	N	1
		Univariate	
		I-square [%]	–
		log DOR; 95%CI	1.55 (0.32; 2.78)
		R; 95%CI	–

Abbreviations: CI, confidence interval; DOR, diagnostic odds ratio.

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
