# Peer review of "Emerging Technologies for Dentin Caries Detection—A Systematic Review and Meta-Analysis"

_jcm, 2022, doi:10.3390/jcm11030674_

Round 1

Reviewer 1 Report

This manuscript is nicely written. However; you must pay attention to the following points:

1-Abstract: the results section is underrepresented and the conclusion is completely missing.

2-Page 2 “aim”: address the aim(s) of the review as a question(s).

3-Materials and Methods: Include the registration number/protocol of the systematic review.

4-Systematic reviews which follow the PRISMA guidelines should address the PICOS, these are missing.

5-Remove the author initials from the main text.

6-Page 5 line 229; explain how did you reach 61 studies in the main text as only those who read the appendix will understand how, and the last paragraph on page 6 isn't sufficiently explanatory.

7-Page 7 (Figure 2): In the risk of bias assessment for in vivo studies (a). The blue color (key) which should stand for unclear risk should be corrected to yellow color according to the colored bars.

8-There is too much referral to the appendices. I suggest including the maximum possible content within the manuscript, this shall avoid problems in reference numbers.

9-Page 19 “Conclusion”: should take into account the results of this meta-analysis.

Reviewer 2 Report

The present manuscript is a systematic review and Meta-Analysis to assess the diagnostic test accuracy of emerging technologies for dentin caries detection.

There are several SR and MA existing on this topic, however, the authors state in the abstract that the paper is dealing with dentin caries detection.  The manuscript can be improved by following some issues:

  1. Titel: The term dentin should be added in the title.
  2. Abstract: line 26: delete the second “of bias”.
  3. Introduction: In this section, the main focus seems to be incipient caries. If the aim is to include dentin caries in the paper, then this aspect should be addressed rather than initial caries lesions.
  4. Throughout the manuscript: The term dentin caries is not clear enough. Do the authors mean cavitated lesion? If yes, the rationale behind using technical aids for detection of cavitated lesions should be discussed. The detection of cavitated lesions is normally performed visually since without further need for additional techniques.
  5. Conclusion: The distinction between the use of devices on occlusal and approximal sites is not clear enough.

Round 2

Reviewer 1 Report

The revised manuscript has improved, indeed, and the authors provided a response to all raised queries. The manuscript can proceed to publication.